# Overexpression of Insulin Receptor Substrate 1 (IRS1) Relates to Poor Prognosis and Promotes Proliferation, Stemness, Migration, and Oxidative Stress Resistance in Cholangiocarcinoma

**DOI:** 10.3390/ijms24032428

**Published:** 2023-01-26

**Authors:** Waleeporn Kaewlert, Chadamas Sakonsinsiri, Worachart Lert-itthiporn, Piti Ungarreevittaya, Chawalit Pairojkul, Somchai Pinlaor, Mariko Murata, Raynoo Thanan

**Affiliations:** 1Department of Biochemistry, Faculty of Medicine, Khon Kaen University, Khon Kaen 40002, Thailand; 2Cholangiocarcinoma Research Institute, Faculty of Medicine, Khon Kaen University, Khon Kaen 40002, Thailand; 3Department of Pathology, Faculty of Medicine, Khon Kaen University, Khon Kaen 40002, Thailand; 4Department of Parasitology, Faculty of Medicine, Khon Kaen University, Khon Kaen 40002, Thailand; 5Department of Environmental and Molecular Medicine, Mie University Graduate School of Medicine, Mie 514-8507, Japan

**Keywords:** cholangiocarcinoma, insulin receptor substrate 1, tumor progression, oxidative stress

## Abstract

Cholangiocarcinoma (CCA) is one of the oxidative stress-driven carcinogenesis through chronic inflammation. Insulin receptor substrate 1 (IRS1), an adaptor protein of insulin signaling pathways, is associated with the progression of many inflammation-related cancers. This study hypothesized that oxidative stress regulates IRS1 expression and that up-regulation of IRS1 induces CCA progression. The localizations of IRS1 and an oxidative stress marker (8-oxodG) were detected in CCA tissues using immunohistochemistry (IHC). The presence of IRS1 in CCA tissues was confirmed using immortal cholangiocyte cells (MMNK1), a long-term oxidative-stress-induced cell line (ox-MMNK1-L), and five CCA cell lines as cell culture models. IRS1 was overexpressed in tumor cells and this was associated with a shorter patient survival time and an increase in 8-oxodG. IRS1 expression was higher in ox-MMNK1-L cells than in MMNK1 cells. Knockdown of IRS1 by siRNA in two CCA cell lines led to inhibition of proliferation, cell cycle progression, migration, invasion, stemness, and oxidative stress resistance properties. Moreover, a transcriptomics study demonstrated that suppressing IRS1 in the KKU-213B CCA cell line reduced the expression levels of several genes and pathways involved in the cellular functions. The findings indicate that IRS1 is a key molecule in the connection between oxidative stress and CCA progression. Therefore, IRS1 and its related genes can be used as prognostic markers and therapeutic targets for CCA therapy.

## 1. Introduction

Cholangiocarcinoma (CCA) is a malignant tumor emerging in the biliary tree. In Northeast Thailand, CCA has the highest mortality and incidence rates in the world [1]. CCA has silent clinical characteristics that make it difficult to diagnose in its early stages and surgical resection is currently the best treatment option [2]. Accumulating evidence suggests that CCA is an oxidative stress-driven carcinoma resulting from chronic inflammation induced by the infection of the liver fluke *Opisthorchis viverrini* [3,4,5]. Oxidative stress is related to cellular biomolecular damage, tissue injury, and altered gene expression levels, all of which are involved in CCA development [6]. Moreover, chronic inflammation caused by *O. viverrini* can trigger the oxidative stress-induced formation of 8-nitroguanine and 8-oxodG in cancer tissues [7]. Recent studies have shown that chronic oxidative stress can induce oncogenesis in cholangiocytes through downregulation of a tumor suppressor gene, early B cell factor 1 (EBF1) [8,9]. However, few studies have been conducted on pathways underlying oxidative stress and their responsive molecules in CCA.

Insulin receptor substrate 1 (IRS1), the first member of the IRS family, is a major adaptor of the insulin receptor and insulin-like growth factor receptor in the insulin signaling pathway. The activation of IRS1 modulates intracellular signaling pathways, including PI3K and MAPK signaling, which are involved in regulation of cellular metabolic and mitogenic pathways [10]. Numerous studies have revealed that IRS1 is implicated in cancer progression. For instance, IRS1 overexpression contributes to cancer progression with the worst clinical outcomes for breast, pancreatic, and prostate cancers through PI3K/Akt and MAPK/ERK signaling pathways [11,12,13,14]. However, there have been few studies on IRS1 function in CCA.

We hypothesized that oxidative stress induces IRS1 expression and that the overexpression of IRS1 contributes to aggressive CCA progression with poor clinical outcomes. To demonstrate that IRS1 is overexpressed in CCA, is related to oxidative stress, and plays a vital role in cancer progression, the expression patterns of IRS1 in CCA tissues were investigated and correlated with 8-oxodG formation and clinicopathological data. To clarify that oxidative stress stimulates IRS1 production, IRS1 expression levels were investigated in a cholangiocyte cell line (MMNK1) and CCA cell lines treated both short- and long-term with hydrogen peroxide (H_2_O_2_). Furthermore, IRS1 knockdown (siIRS1) CCA cell lines were used to assess cellular function related to cell proliferation, migration, invasion, cell cycle arrest, stemness, and oxidative stress resistance properties. IRS1-related genes and pathways were identified using the RNA sequencing technique, Reactome pathway enrichment analysis, and real-time PCR technique in the IRS1 knockdown CCA cells.

## 2. Results

### 2.1. IRS1 Expression Profiles in CCA Patients in Relation to Clinicopathological Features and 8-Oxodg Formation

IRS1 was predominantly localized in the cytoplasm and nucleus of CCA cells, with slight expression in adjacent normal bile ducts and hepatocytes (Figure 1). Expression of IRS1 was found to be high in 63% (53/84) of 84 cases of CCA, but low in the remaining 37% (31/84). The IRS1 expression profile showed no correlation with age, gender, histological type, and metastasis status (Table 1). IRS1 expression demonstrated a positive correlation with 8-oxodG formation (Table 1). High IRS1 expression was significantly correlated with short survival time in CCA patients (Figure 2A). Moreover, the combination of high IRS1 expression and high 8-oxodG formation was significantly correlated with high metastasis status (Table 2) and the lowest survival rate compared to individuals in other groups (Figure 2B). These results suggested that oxidative stress may lead to aggressive CCA progression with poor clinical outcomes by activating IRS1.

### 2.2. Expression of IRS1 in Cells Exposed Long-Term to H_2_O_2_

To prove that oxidative stress can induce IRS1 expression, IRS1 expression levels were measured in MMNK1, H_2_O_2_-treated MMNK1, and CCA cells. The ox-MMNK1-L cell line was established by daily treating MMNK1 cells for an extended period of time (72 days) with 25 μM of H_2_O_2_ [8]. Cell proliferation and migration rates of the oxidative-stress-resistant cells (ox-MMNK1-L) were significantly higher than the parental cells (MMNK1) (Appendix A). Moreover, the ox-MMNK1-L cells showed increased IRS1 expression at both the mRNA and protein levels compared to the parental cells, as shown in Figure 3. However, short-term treatment with H_2_O_2_ (0, 10, 25, 50, and 100 μM) for 24, 48, and 72 h had no effect on IRS1 expression levels in MMNK1 and CCA cell lines (Appendix A).

### 2.3. Transient IRS1 Knockdown in CCA Cell Lines

To demonstrate that IRS1 expression levels play critical roles in CCA progression, IRS1 knockdown CCA cell lines were used. Expression of IRS1 at the mRNA and protein levels was observed in MMNK1 and five CCA cell lines (KKU-100, KKU-023, KKU-055, KKU-213A, and KKU-213B). The mRNA levels of IRS1 were high in KKU-023, KKU-055, KKU-213A, and KKU-213B cells when compared with MMNK1 cells (Figure 4A). IRS1 was localized in the cytoplasm and nucleus of all cell lines and its protein level was elevated in CCA cell lines compared to MMNK1 (Figure 4B,C). To elaborate the effect of IRS1 on the progression of CCA, the highly aggressive cancer cell lines with high expression of nuclear IRS1 and cytoplasmic IRS1 (KKU-213A and KKU-213B, respectively) were used for siRNA-mediated transient IRS1 knockdown. These cell lines exhibit differences in morphologies, phenotypes, and gene expression profiles as well as tumor formation in xenograft models [15]. After 48 h (KKU-213A) and 72 h (KKU-213B) of siRNA transfection, the IRS1 mRNA level was significantly down-regulated (Figure 4D), and total IRS1 protein as well as nuclear and cytosolic IRS1 protein expressions were diminished in the knockdown cells compared to those of the scramble controls (Figure 4E,F).

### 2.4. Transient IRS1 Knockdown Suppressed Proliferation of CCA Cell Lines

The IRS1 knockdown cells were subjected to functional analysis. As shown in Figure 5A,B, the IRS1 knockdown could reduce the viability of KKU-213A and KKU-213B cells. Moreover, the RNA sequencing results demonstrated the absolute fold changes of cell proliferation, and DNA replication-related genes were down-regulated in the IRS1 knockdown CCA cells (siIRS1) compared to the vehicle control cells (scramble), as shown in Figure 5C. The pathways related to cell proliferation and DNA replication were also verified using the Reactome database (Figure 5D). We further validated the expression profiles of interesting genes from RNA-Seq results using real-time PCR. The results demonstrated that an anti-apoptosis gene (baculoviral IAP repeat containing 5; BIRC5) and oncogenic genes, including yes-associated protein 1 (YAP1) and mitogen-activated protein kinase kinase kinase 3 (MAP3K3), were significantly decreased after IRS1 knockdown in KKU-213A and KKU-213B cells (Figure 5E,F).

### 2.5. Transient IRS1 Knockdown Induced G1 Cell Cycle Arrest of CCA Cell Lines

Considering that cell proliferation is directly associated with cell cycle progression, flow cytometry with propidium iodide staining was used to investigate the cell cycle distribution in IRS1 knockdown CCA cells (Appendix A). The results demonstrated that siIRS1 could induce G1 phase cell cycle arrest in both cell lines. KKU-213A cells in the G1 phase increased from 68.57 to 75.10%, whereas KKU-213B cells in the G1 phase increased from 74.43 to 89.27% (Figure 6A,B). RNA sequencing analysis demonstrated that many genes (e.g., CCNE1 and CDK2) that regulate cell cycle progression were significantly decreased in the IRS1 knockdown CCA cell lines (Figure 6C). Many of the cell cycle-progression-regulated pathways were also shown to be reduced in the knockdown cell lines (Figure 6D). These results indicated that IRS1 can promote cell cycle progression from G1 to S phases, resulting in increased CCA cell proliferation.

### 2.6. Transient IRS1 Knockdown Reduced Stem-like Properties of CCA Cell Lines

The role of IRS1 on stem cell-like properties of CCA cell lines was determined by spheroid culture using the hanging drop method. As shown in Figure 7A,B, the spheroid formation ability of CCA cell lines was significantly decreased after IRS1 knockdown. The RNA sequencing results showed that mRNA levels of stem cell-associated genes and pathways were decreased in the siIRS1 condition (Figure 7C,D). Additionally, the expressions of stem cell markers i.e., krueppel-like factor 4 (KLF4) and spalt-like transcription factor 4 (SALL4), were radiated using the quantitative RT-PCR (Figure 7E,F). In KKU-213A and KKU-213B, the knocking down of IRS1 suppressed SALL4 expression levels. Taken together, these results support the role of IRS1 in CCA progression through activation of the stem cell-associated gene (SALL4) and stem cell properties.

### 2.7. Transient IRS1 Knockdown Decreased Migration and Invasion Activities of CCA Cell Lines

To examine the role of IRS1 on migration and invasion of CCA cell lines, transwell Boyden chamber migration and invasion assays were performed in IRS1 knockdown cells. The percentages of migrated KKU-213A and KKU-213B cells were lower after IRS1 knockdown at 18 h post-migration (Figure 8A,B). Likewise, siIRS1 could suppress the invasion of both KKU-213A and KKU-213B CCA cell lines at 18 h when compared with the scramble condition (Figure 8C,D). Furthermore, mRNA levels of many epithelial–mesenchymal transition (EMT)-related genes were significantly reduced in IRS1 knockdown CCA cell lines analyzed by RNA-Seq (Figure 8E). The TGF-*β* signaling in EMT and negative regulation of mesenchymal–epithelial transition (MET) activities were significantly reduced in the knockdown cells analyzed by Reactome pathway analysis (Figure 8F). In addition, the quantitative RT-PCR revealed that transforming growth factor-*β* receptor 1 (TGFBR1) and vimentin (VIM) were also significantly decreased in the knockdown cell lines (Figure 8G,H). These findings demonstrated that IRS1 plays a pivotal role in CCA progression by promoting migration and invasion activities.

### 2.8. Transient IRS1 Knockdown Reduced Oxidative Stress-Resistant Property of CCA Cell Lines

As CCA is one of the cancers associated with oxidative stress [6], we investigated the association between oxidative stress and IRS1 expression in CCA. The IRS1 knockdown CCA cell lines (KKU-213A and KKU-213B) were treated with various concentrations of H_2_O_2_ for 48 h, and cell viability was measured by the MTT assay. Significantly fewer viable cells were present in the IRS1 knockdown treatment relative to the scramble control cells at 200 μM and higher concentrations of H_2_O_2_ (Figure 9A,B). The oxidative stress biomarker (carbonylated protein) levels were also increased in the knockdown cell lines (Figure 9C), suggesting that a lack of IRS1 expression increases oxidative stress. Many of the DNA repairing systems were significantly reduced in the knockdown cells analyzed by Reactome pathway analysis (Figure 9D). The oxidative stress-resistant-related genes (antioxidant and DNA repairing genes) in the knockdown cells are shown in Figure 9E. The expression levels of nuclear factor erythroid 2-related factor 2 (NRF2), superoxide dismutase-2 (SOD2), and catalase (CAT) were also measured using real-time PCR. NRF2 expression was significantly decreased in IRS1 knockdown KKU-213A cells and trended lower in IRS1 knockdown KKU-213B cells when compared to controls. The silencing of IRS1 promoted SOD2 expression in both CCA cell lines but had no effect on CAT expression (Figure 9F,G). This finding suggested that the suppression of IRS1 increased ROS generation and DNA damage through down-regulations of NRF2, and its downstream genes as well as DNA repairing systems led to inhibition of the oxidative stress resistance property of the cancer cells.

## 3. Discussion

IRS1 is a cytoplasmic adaptor protein that transmits insulin and insulin-like growth factor receptor signals in the insulin signaling pathway, which controls various cellular processes [10]. IRS1 has been discovered in the nucleus of cancer cells, where it plays synergistic roles with IRS1 in the cytoplasm to stimulate cell cycle progression, DNA repair, cell proliferation, and estrogen responsiveness [16,17,18]. IRS1 has been linked to the development and progression of various cancers, including breast, prostate, and pancreatic cancers [11,12,13,14]. IRS1 expression is elevated in breast invasive ductal carcinoma and associated with p-Akt expression and a poor prognosis for the patients [11]. Meng et al. demonstrated that the inhibition of IRS1 expression reduces proliferation of prostate cancer cells, and inhibits cell cycle progression and ERK activation [13]. Our results also showed that IRS1 expression was elevated in CCA and significantly correlated with a poor prognosis. IRS1 knockdown suppressed cell proliferation, migration, invasion, and induced G1-phase cell cycle arrest in CCA cells. In addition, the transcriptomic analysis confirmed that the suppression of IRS1 in CCA cells could down-regulate many gene clusters involved in a variety of biological functions, e.g., DNA replication, DNA repair, cell cycle, TGF-β signaling, and NOTCH signaling pathways. These findings support the significance of IRS1 in the development and progression of CCA with the most aggressive phenotypes.

The serine/threonine kinase MAP3K3, also known as MEKK3, is a member of the MAP3K superfamily. MEKK3 is expressed in several cell types and regulates many cellular processes, such as cell proliferation, survival, differentiation, apoptosis, and development via different signaling pathways [19]. MEKK3 plays important roles in cancer initiation and progression through different downstream signaling pathways, such as NFκB, YAP/TAZ, and AKT [20,21,22]. In pancreatic cancer, MEKK3 induces cancer stemness and aggressiveness by altering the transcriptional activities of YAP/TAZ [20]. YAP1, a transcriptional regulator, is a critical downstream effector of the Hippo pathway in controlling cell fate, proliferation, and apoptosis [23]. Particularly, elevated YAP1 has been reported in CCA and has been shown to induce CCA carcinogenesis and metastasis via AKT pathway activation [24]. Furthermore, IRS1 regulates the nuclear localization of YAP1, which causes the proliferation of cerebellar neural precursors in hedgehog-associated medulloblastomas [25]. The present study has shown that the silencing of IRS1 down-regulated MAP3K3, YAP1, and BIRC5 mRNA expression levels. BIRC5 is an anti-apoptotic gene implicated in the development and apoptosis resistance of CCA [26]. The expression of BIRC5 is up-regulated by the YAP1 in esophageal and colorectal cancer [27,28]. These results imply that IRS1 promoted proliferation of CCA cells, metastasis, and stemness by up-regulating the expression of MAP3K3 and YAP1.

Up-regulated IRS1 has been associated with differentiation and tumorigenesis, and the Wnt/β-catenin signaling pathway [29,30]. The present results show that suppression of IRS1 repressed stem cell properties of CCA cells through down-regulation of NOTCH, telomere maintenance, and SUMOylation signaling pathways. Furthermore, the RNA-Seq results demonstrated that stem cell markers, i.e., KLF4 and SALL4, were down-regulated in IRS1 knockdown KKU-213B cells. SALL4 was successfully validated in the two CCA cell lines using real-time PCR. SALL4 is also involved in the maintenance of embryonic stem cell pluripotency and early embryonic development [31]. Zhu et al. demonstrated that increased SALL4 was associated with poor prognosis in CCA patients and induced malignant phenotypes of CCA by regulating PTEN/PI3K/Akt and Wnt/β-catenin signaling pathways and promoting the EMT process [32]. We found that the expression level of SALL4 was decreased in IRS1-silenced CCA cells. These findings indicate that IRS1 stimulated expression of SALL4, which increased the stemness and aggressiveness of CCA cells, leading to tumor progression.

The present results showed that IRS1 also plays a vital role in CCA invasion and migration. High IRS1 expression tended to be correlated with metastasis status of CCA patients. The suppression of IRS1 could inhibit the expression of many genes in the negative regulation of MET activity and TGF-β receptor signaling in EMT pathways including LRIG1, PTPN1, RHOA, RPS27A, TGFBR1, UBA52, UBB, and VIM. Additionally, the high level of IRS1 induced the transcription of TGF-β receptor 1 and vimentin in both CCA cell lines. TGF-β receptor 1 has been reported in several cancers and serves as an important receptor in TGF-β1 signaling, causing the EMT pathway and resulting in cell migration and invasion [33]. In CCA, TGF-β1 induces EMT and expression of mesenchymal markers (N-cadherin, vimentin, and fibronectin), leading to increases in cell migration and invasion [34]. These reports all support the hypothesis that IRS1 plays an important role in CCA metastasis through the up-regulation of TGF-β1, which, in turn, induces the EMT signaling pathway.

Persistent oxidative stress, induced by chronic inflammation, is associated with numerous inflammation-related cancers, including CCA. Chronic oxidative stress is known to alter gene expression and cause biomolecular damage in oxidative stress-driven CCA genesis [3,5,8,35]. In general, high oxidative stress induces cellular damage via oxidative damage to biomolecules and, consequently, cell death, while low oxidative stress promotes cell growth and adaptation. Prolonged exposure of MMNK1 cells to low concentrations of H_2_O_2_ causes the up-regulation of antioxidant genes, leading to an increase in oxidative stress resistance properties [8]. Persistent oxidative stress induces CCA tumorigenic properties by suppression of EBF1 expression and activation of ZNF423 expression, resulting in CCA progression with poor prognosis [9,36,37]. Our findings also indicated a strong correlation between IRS1 expression levels and oxidative stress. In addition, compared to those in other groups, CCA patients with high IRS1 expression and high oxidative stress had the lowest survival rates and the highest metastatic status. These findings drive the hypothesis that IRS1 activation in CCA likely caused by oxidative stress leads to CCA progression with aggressive clinical outcomes. Furthermore, IRS1 expression in the cholangiocyte cells could be induced by the persistence of oxidative stress. Moreover, the IRS1 knockdown CCA cells had lower resistance against oxidative stress because of decreased expressions of many genes involved in DNA repairing and antioxidant systems with increased SOD2 expression, but CAT expression was unchanged relative to controls. All of these may contribute to increased ROS generation through H_2_O_2_ production and Fenton’s reaction, both of which result in high oxidative stress that can be detected by the increase in carbonylated proteins in the IRS1 knockdown CCA cell lines. In combination, chronic inflammation induces persistent oxidative stress that up-regulates IRS1, which also inhibits ROS generation and protects cellular damage via up-regulating NRF2 and its downstream antioxidant genes as well as the DNA repairing systems, thereby enhancing the ability of the cells to survive under oxidative stress. This adaptation, and the selective clonal expansion of these resistant cells, would be involved in oxidative stress-driven CCA genesis.

In general, the actions of IRS1 are controlled by post-translational modifications, i.e., tyrosine and serine phosphorylation. The distinct phosphorylation sites of IRS1 mediate positive or negative regulation of IRS1, which are induced by different upstream regulators such as insulin receptor, insulin like growth factor 1, cytokines, and hormones [38,39]. There are many studies that have reported that short-term exposure to oxidative stress induces serine phosphorylation of IRS1 and insulin resistance in insulin-dependent cells [40,41]. In hepatoma cells, ROS inhibit IRS1 function in insulin signaling through the induction of IRS1 phosphorylation on Ser307 [42]. On the other hand, the present study showed that the exposure of low H_2_O_2_ concentration (25 µM) for a long time (72 days) to non-insulin-dependent cells (cholangiocytes, MMNK1) increased IRS1 expression compared with the parent cells. These contradictory results may be due to many factors including the different model, cell types, condition, and the duration of response. The results from the acute response of the insulin-dependent cells and our results from the long-term response to oxidative stress (cell adaptation) in insulin-independent cells are not comparable. Interestingly, Chan et al. demonstrated that IRS1-overexpressing NIH/3T3 cells have a high cell proliferation rate with resistance to oxidative stress [43]. These results supported our findings that CCA cells overexpressing IRS1 showed high cell proliferation and oxidative stress-resistant abilities.

## 4. Materials and Methods

### 4.1. Human Intrahepatic CCA Tissues

Paraffin-embedded CCA tissues (*n* = 84; 60 males and 24 females) and clinical data were acquired from the Cholangiocarcinoma Research Institute, Faculty of Medicine, Khon Kaen University, Khon Kaen, Thailand. The protocols for tissue collection and experimental design were approved by the Center for Ethics in Human Research, Khon Kaen University (HE571283 and HE641520).

### 4.2. Immunohistochemistry

Formalin-fixed paraffin-embedded CCA tissues were deparaffinized and rehydrated. Heat-induced antigen retrieval was performed using a pressure cooker (5 min in 10 mM sodium citrate buffer, pH 6, with 0.5% Tween-20). Endogenous peroxidase activity and non-specific binding were blocked with 0.3% (v/v) H_2_O_2_ in phosphate-buffered saline (PBS), followed by 10% skim milk in PBS. Rabbit anti-IRS1 (1:100; SAB4300482, Sigma-Aldrich Crop, Saint Louis, MO, USA) was used as the primary antibody. Peroxidase-conjugated secondary antibody was subsequently added. The peroxidase activity was determined using a DAB (3,3′-diaminobenzidine) substrate kit (Vector Laboratories, Inc., Burlingame, CA, USA). Counterstaining was performed using Mayer’s hematoxylin, and stained tissues were observed under a light microscope. The IHC score (0–12) was calculated by multiplying the intensity score (defined as 0–3 for none, weak, moderate, and strong, respectively) by the frequency score (defined as 0 = none, 1 = 1–25%, 2 = 26–50%, 3 = 51–75%, and 4 ≥ 76%) [44]. The mean value of the IHC score was used as the cut-off for high/low expression. IHC results for the formation of 8-oxodG were obtained from our previous study [9].

### 4.3. Cell Culture

Five CCA cell lines, KKU-023, KKU-055, KKU-100, KKU-213A, and KKU-213B, were derived from primary tumors surgically removed from CCA patients in the Srinagarind Hospital, Khon Kaen University, Thailand. All cell lines were obtained from the Cholangiocarcinoma Research Institute, Khon Kaen University, Thailand. An immortal cholangiocyte cell line, MMNK1, was established by Maruyama et al. [45]. An oxidative stress-resistant cell line (ox-MMNK1-L) was established from MMNK1 via prolonged exposure to H_2_O_2_ [8]. All cell lines were cultured in Ham’s F-12 medium supplemented with 10% fetal bovine serum and 100 mg/mL of penicillin streptomycin (both from Gibco/Life Technologies, Grand Island, NY, USA). Cells were grown in a humidified incubator containing 5% CO_2_ at 37 °C.

### 4.4. Real-Time PCR

Total RNA was isolated from cells using TRIzol™ reagent (Invitrogen, Carlsbad, CA, USA). The High-Capacity cDNA Reverse Transcription kit (Applied Biosystems, Foster, CA, USA) was utilized for cDNA synthesis. Gene expression was detected using Taqman gene expression assays (IRS1; Hs00178563_m1, BIRC5; Hs04194392_s1, NRF2; Hs00975960_m1, MAP3K3; Hs00176747_m1, YAP1; Hs00902712_g1, CAT; Hs00156308_m1, SOD2; Hs01553554_m1, KLF4; Hs00358836_m1, SALL4; Hs00360675_m1, TGFBR1; Hs00610320_m1, VIM; Hs00958111_m1, β-actin; Hs99999903_m1, Applied Biosystems, Foster, CA, USA) with an ABI QuantStudio™ 6 Flex Real-Time PCR System. The relative mRNA expression was determined using the delta-delta Ct method with β-actin as an internal control.

### 4.5. Immunocytochemistry

Cell lines (5 − 7 × 10^4^ cells) were cultured overnight in a 48-well plate with complete Ham’s F-12 medium to allow cell attachment. The cells were fixed with 4% (*w*/*v*) paraformaldehyde in PBS and incubated with 0.2% (*v*/*v*) Triton X-100 in PBS for the purpose of decreasing cell permeabilization. The 0.3% (*v*/*v*) H_2_O_2_ solution and 3% (*w*/*v*) bovine serum albumin in PBS were used to block endogenous peroxidase and non-specific binding, respectively. As the primary antibody, rabbit anti-IRS1 (1:300; SAB4300482, Sigma-Aldrich Crop, Saint Louis, MO, USA) was used, along with a peroxidase-conjugated secondary antibody. DAB substrate kit (Vector Laboratories, Inc., Burlingame, CA, USA) was utilized to develop signals. The stained cells were observed under an inverted microscope.

### 4.6. Western Blot Analysis

Protein was extracted from cells using the radioimmunoprecipitation lysis (RIPA) buffer and protein concentration was measured using the Bradford assay (Bio-Rad Laboratories, Inc., Hercules, CA, USA). Protein samples (20 μg) were separated by 10% sodium dodecyl sulfate–polyacrylamide gel electrophoresis and then transferred onto polyvinylidene fluoride membranes (EMD Millipore, Billerica, MA, USA). Non-specific binding was prevented using 5% (*w*/*v*) skim milk in Tris-buffered saline containing 0.1% (*v*/*v*) Tween-20. The membranes were probed with the specific primary antibody: rabbit anti-IRS1 (1:2500 dilution; 17509-1-AP, Proteintech Group Inc., Rosemont, IL, USA), and subsequently incubated with horseradish peroxidase-conjugated secondary antibody. Enhanced chemiluminescence reagent was used to develop the signals, which were then analyzed using the Amersham ImageQuant 800 (Cytiva Life Sciences™, Marlborough, MA, USA). The IRS1 protein band was seen at approximately 120 kDa, and β-actin, which served as an internal control, was detected at approximately 42 kDa.

### 4.7. Small Interfering RNA (siRNA) Transfection

The highly aggressive CCA cell lines with a high expression of nuclear IRS1 and cytoplasmic IRS1 (KKU-213A and KKU-213B, respectively) were used for siRNA-mediated transient IRS1 knockdown. ON-TARGETplus SMARTpool Human IRS1 (siIRS1; Catalog ID: L-003015-00-0005), which provides a mixture of 4 siRNA sequences in a single reagent, was used for IRS1 transient knockdown in conjunction with ON-TARGETplus Non-targeting Control siRNA as a control (scramble; Dharmacon™, Lafayette, CO, USA). In 6-well plates, cells (6.0 × 10^4^ cells) were transfected with Lipofectamine™ RNAiMax reagent (Invitrogen, Carlsbad, CA, USA) according to the manufacturer’s instructions. Cells were harvested at 48 or 72 h after transfection for the functional analysis.

### 4.8. Hydrogen Peroxide Treatment

Cells were reseeded at a density of 3 × 10^3^ cells/well in 96-well plates and cultured for 24 h. The cells were treated with 0, 10, 25, 50, 100, 200, 300, 400, and 500 μM of H_2_O_2_ in complete medium and re-treated at 24 h. After 48 h, cells were subjected to the MTT assay to determine whether IRS1 is implicated in oxidative stress resistance.

### 4.9. Carbonylated Protein Determination

The measurement of protein carbonyl content was performed using the Protein Carbonyl Content Assay Kit (Merck KGaA, Darmstadt, Germany) following the manufacturer’s protocol. The protein samples (~100 μg) were extracted from IRS1 knockdown cells using RIPA. Then, 100 μL of each sample was treated with 10% streptozocin solution to decrease nucleic acid interference. The samples were subjected to the 2,4-dinitrophenylhydrazine (DNPH) assay to determine the carbonyl content. After the reaction, any protein carbonyl-dinitrophenyl hydrazone formed was dissolved in 6 M guanidine solution and the absorbance measured at 375 nm. The amount of protein in each sample was determined using the bicinchoninic acid assay and measured spectrophotometrically at 562 nm. Then, the protein carbonyl content was calculated and represented as nmol of carbonylated protein per mg of protein.

### 4.10. Cell Proliferation Assay

Cell proliferation was determined using the MTT assay. Cells were trypsinized and plated into a 96-well plate at a density of 3 × 10^3^ cells/well. After 6, 24, 48, and 72 h of incubation, the cells were washed with PBS prior to the addition of 100 μL of MTT reagent (0.5 mg/mL) (Sigma-Aldrich Crop, Saint Louis, MO, USA). Dimethyl sulfoxide was used to dissolve the purple-colored formazan product. The absorbance at 540 nm was determined using a microplate reader (Tecan Group Ltd., Männedorf, Switzerland).

### 4.11. Cell Migration Assay

Transwell chambers with 8 μm pore-size filters (Corning Inc., Corning, NY, USA) were used for cell migration assay. After IRS1 knockdown, KKU-213A (2 × 10^4^ cells) and KKU-213B (5 × 10^4^ cells) in serum-free medium were added to the upper chambers, while the lower chambers were filled with complete medium containing 10% fetal bovine serum. After 18 h of incubation, cells were allowed to migrate through the underside of the porous filters. The migrating cells were fixed with absolute methanol and stained with Mayer’s hematoxylin. The stained cells were counted under a microscope.

### 4.12. Cell Invasion Assay

Transwell chambers with an 8 μm pore-size membrane coated with 0.4 mg/mL of Matrigel™ (Corning Inc., Corning, NY, USA) were used for the invasion assay. The IRS1 knockdown cells (3 × 10^4^ of KKU-213A cells and 5 × 10^4^ of KKU-213B cells) in serum-free medium were added to the upper chambers and the lower chambers were filled with complete medium. After 18 h of incubation, cells were allowed to invade through the extracellular matrix and attached to the other side of porous filters. The invasive cells were fixed with absolute methanol, stained with Mayer’s hematoxylin, and counted under a microscope.

### 4.13. Cell Cycle Arrest Assay

Cell cycle analysis was investigated using flow cytometry following propidium iodide staining. Cells were harvested at 48 or 72 h post-transfection and washed with PBS. Cells were fixed in 70% ethanol at 4 °C overnight. The fixed cells were incubated with 30 μg/mL of RNase A solution (QIAGEN Sciences, Inc., Germantown, MD, USA) for 30 min at 37 °C, and 20 μg/mL of propidium iodide solution (EMD Millipore, Billerica, MA, USA) was subsequently added. Cell cycle distribution was analyzed using a Becton Dickinson FACSCanto II flow cytometer and FACSDiva software version 6.1.3 (BD Biosciences, San Jose, CA, USA).

### 4.14. RNA Sequencing

Total RNA from IRS1 knockdown cells was extracted using a NucleoSpin^®^ RNA Plus kit (Macherey-Nagel GmbH & Co. KG, Düren, Germany) according to the manufacturer’s protocol. The quantity and quality of RNA were assessed with a NanoDrop ND-2000 spectrophotometer (NanoDrop Technologies, Wilmington, DE, USA) and Agilent 2100 Bioanalyzer system (Agilent Technologies, Inc., Santa Clara, CA, USA). RNA samples were subjected to whole-transcriptome sequencing using paired-end sequencing (2 × 150 bp) on a NovaSeq high-throughput sequencer (Illumina, Inc., San Diego, CA, USA) at Molecular Genomics Pte. Ltd., Singapore. Alignment was performed using the COBWeb algorithm implemented in Strand NGS (Agilent Technologies, Inc., Santa Clara, CA, USA). Trimmed readings were aligned with default parameters to the hg38 and RefSeq genes and transcripts. Normalization was achieved using DESeq and all samples were baselined to the median. Fold change > 1.2 and *p*-value < 0.05 were used as cut-off values for differentially expressed genes. Genes that were differentially expressed were validated with DAVID Bioinformatics Resources (https://david-d.ncifcrf.gov/home.jsp, accessed on 30 September 2022) for gene ontology and the Reactome pathway [46].

### 4.15. Spheroid Formation

Ten microliters of cell suspension (5000 cells) in complete medium was hanging-drop-cultured on a cell culture dish and incubated for 24–48 h in a humidified incubator with 5% CO_2_ at 37 °C. An inverted microscope was used for reviewing spheroid formation. The spheroid formation was adapted from Foty R. [47].

### 4.16. Data analysis

All assays were performed in three independent experiments and the results were presented as mean ± SD. The SPSS Statistics software version 28.0 (IBM Corporation, Armonk, NY, USA) and GraphPad Prism 9 software (GraphPad Software, Inc., San Diego, CA, USA) were used for statistical analysis. The survival analysis of patients was assessed using the Kaplan–Meier estimate with log-rank test. Pearson’s chi-square test was used to analyze the correlation between protein expression patterns and clinicopathological data of the patients. Student’s t-test was applied to compare between two groups. * *p*-value < 0.05 was considered as statistically significant.

## 5. Conclusions

According to our present findings, the mechanism of IRS1 in CCA genesis is depicted in Figure 10. IRS1 is activated by chronic inflammation-induced persistent oxidative stress. IRS1 induces CCA cell proliferation, migration, invasion, and cell cycle progression through stimulation of various genes involved in cellular processes, including the cell cycle, DNA replication, TGF-β signaling, NOTCH signaling pathway, and activation of oncogenic gene expression, e.g., MAP3K3, YAP1, BIRC5, TGFBR1, and VIM. In addition, IRS1 increases expressions of NRF2 and its downstream (GPX2, GSTA1, IDH1, and PHGDH) as well as DNA repair pathways, resulting in oxidative stress resistance in CCA cells. Furthermore, IRS1 promotes the transcription of the tumor stem cell marker SALL4, suggesting that it might be implicated in the stemness property of CCA cells. Taken together, IRS1 performs an oncogenic function in CCA, which contributes to the aggressive progression of the disease. Thus, IRS1 and its related genes could serve as biomolecular markers and therapeutic targets for treatment of CCA patients.

## Figures and Tables

**Figure 1 ijms-24-02428-f001:**
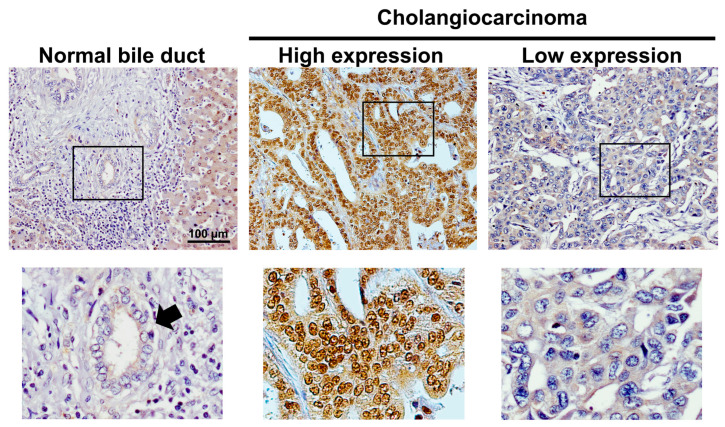
Expression patterns of IRS1 in normal bile ducts and CCA tissues determined using immunohistochemistry. All figures on the upper panel were taken at 100× magnification. Squares represented areas of the top row that were enlarged in the bottom row. Arrow indicates normal bile duct.

**Figure 2 ijms-24-02428-f002:**
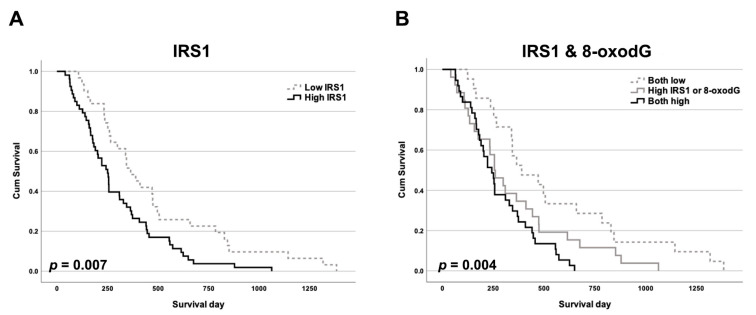
The Kaplan–Meier survival curves of patients against IRS1 and 8-oxodG levels. Impact on overall survival of CCA patients (*n* = 84) of (**A**) IRS1 expression and (**B**) combined IRS1 expression and 8-oxodG formation.

**Figure 3 ijms-24-02428-f003:**
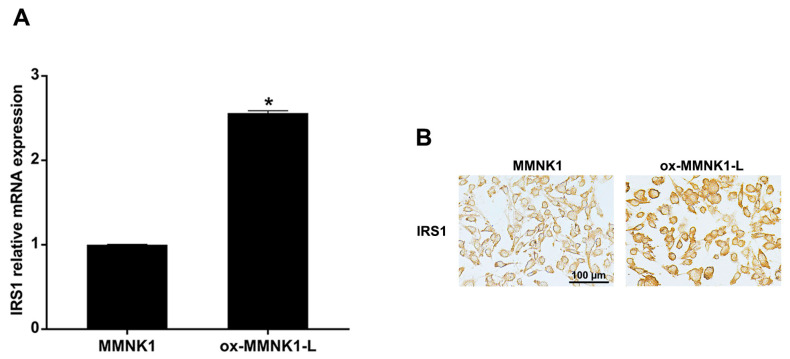
Expression of IRS1 in an oxidative-stress-resistant cholangiocyte cell line (ox-MMNK1-L). (**A**) Relative IRS1 mRNA expression levels in MMNK1 and ox-MMNK1-L cells assessed by real-time PCR. * *p*-value < 0.05 compared with MMNK1. (**B**) Immunocytochemical staining of IRS1 in ox-MMNK1-L and MMNK1 cells.

**Figure 4 ijms-24-02428-f004:**
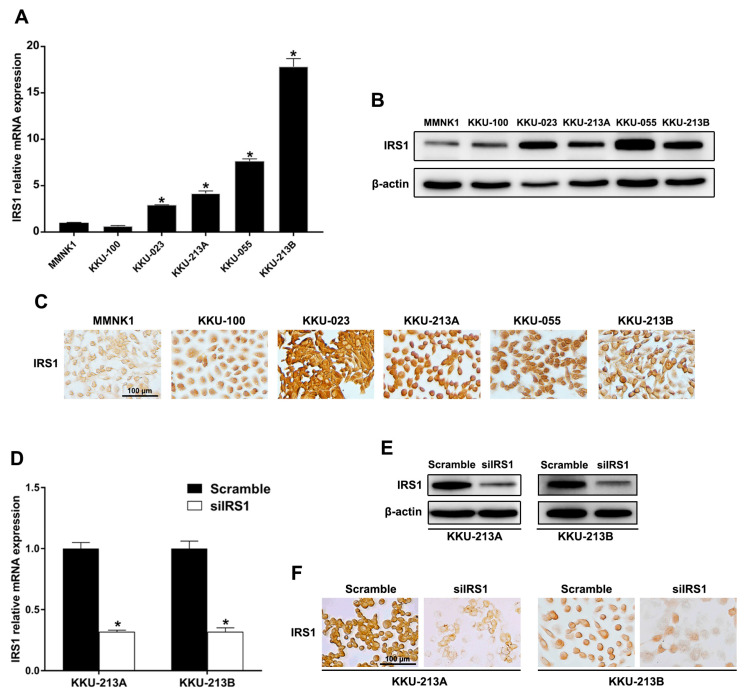
IRS1 expression in human CCA and immortalized cholangiocyte cell lines. (**A**) Relative mRNA levels of IRS1 measured by real-time PCR. * *p*-value < 0.05 compared to MMNK1. (**B**) Western blot analysis of IRS1 protein. (**C**) Immunocytochemical staining of IRS1 protein. (**D**) Relative mRNA expression of IRS1 in IRS1 knockdown CCA cells (KKU-213A and KKU-213B). * *p*-value < 0.05 compared to scramble. (**E**,**F**) IRS1 protein expression in KKU-213A and KKU-213B cells after IRS1 knockdown detected by Western blotting and immunocytochemistry, respectively. Original magnification was 200× for all micrographs.

**Figure 5 ijms-24-02428-f005:**
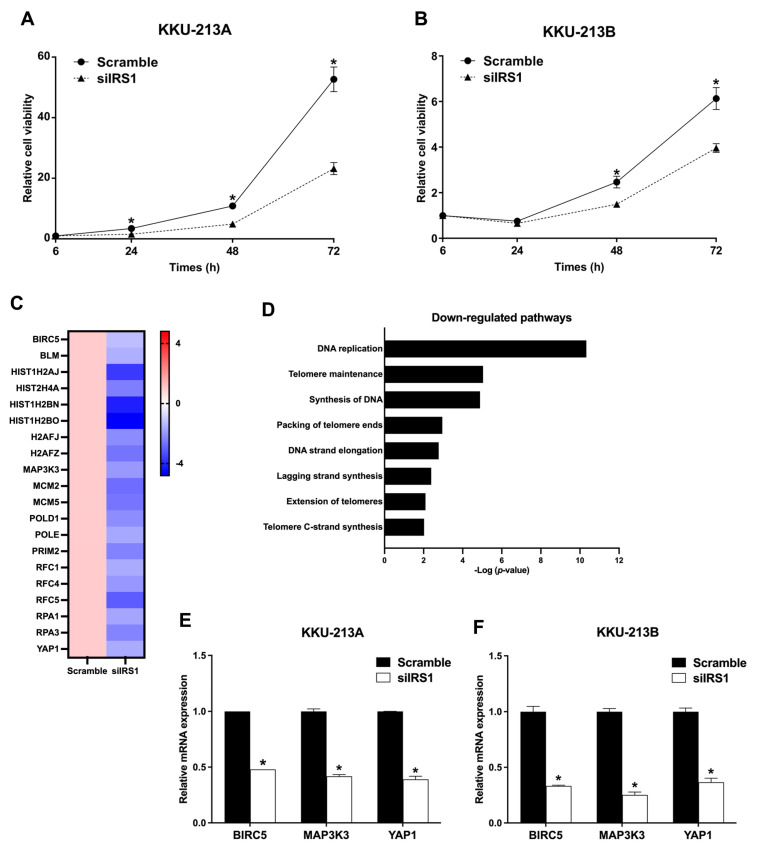
Effect of IRS1 knockdown on proliferation of CCA cell lines. (**A**,**B**) Effect of IRS1 knockdown on cell viability of KKU-213A and KKU-213B determined by MTT assay. (**C**) Heat map representing fold changes of cell proliferation-related genes in KKU-213B after IRS1 knockdown investigated by RNA sequencing. (**D**) Reactome pathway enrichment analysis revealed that many proliferation-related pathways were down-regulated in IRS1 knockdown cells. (**E**,**F**) Relative mRNA expression of BIRC5, MAP3K3, and YAP1 in IRS1 knockdown CCA cells. * *p*-value < 0.05 compared to scramble.

**Figure 6 ijms-24-02428-f006:**
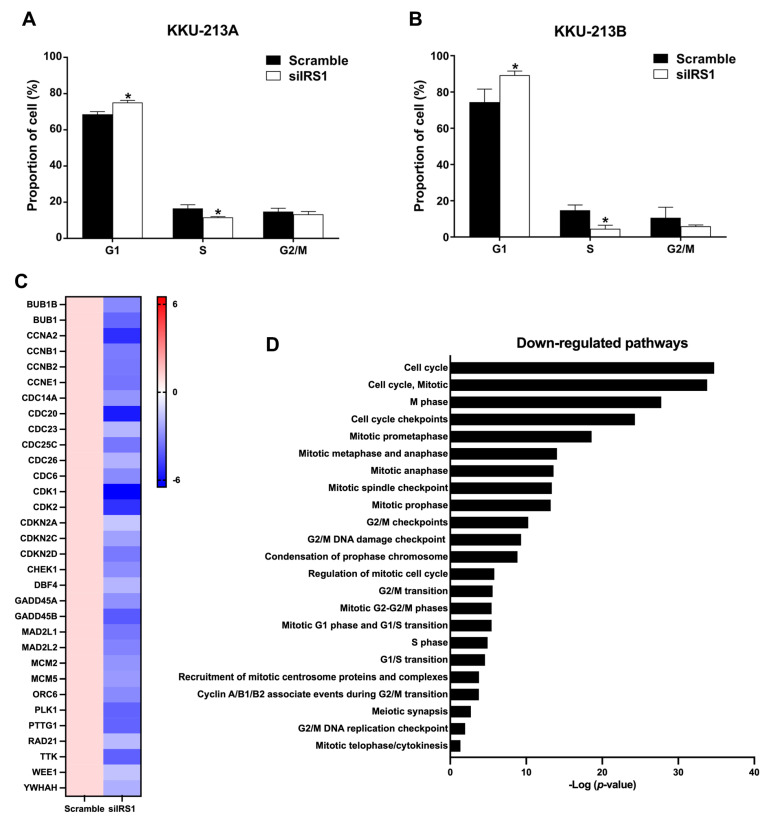
Effect of IRS1 knockdown on cell cycle progression of CCA cell lines. (**A**,**B**) Percentage of cells at different cell cycle stages after IRS1 knockdown and propidium iodide staining, as detected by flow cytometry. (**C**) Heatmap indicates fold changes of cell cycle-related gene expression in IRS1-silenced KKU-213B cells investigated by RNA sequencing. (**D**) Down-regulated pathways in IRS1 knockdown cells analyzed by Reactome pathway enrichment. * *p*-value < 0.05 compared to scramble.

**Figure 7 ijms-24-02428-f007:**
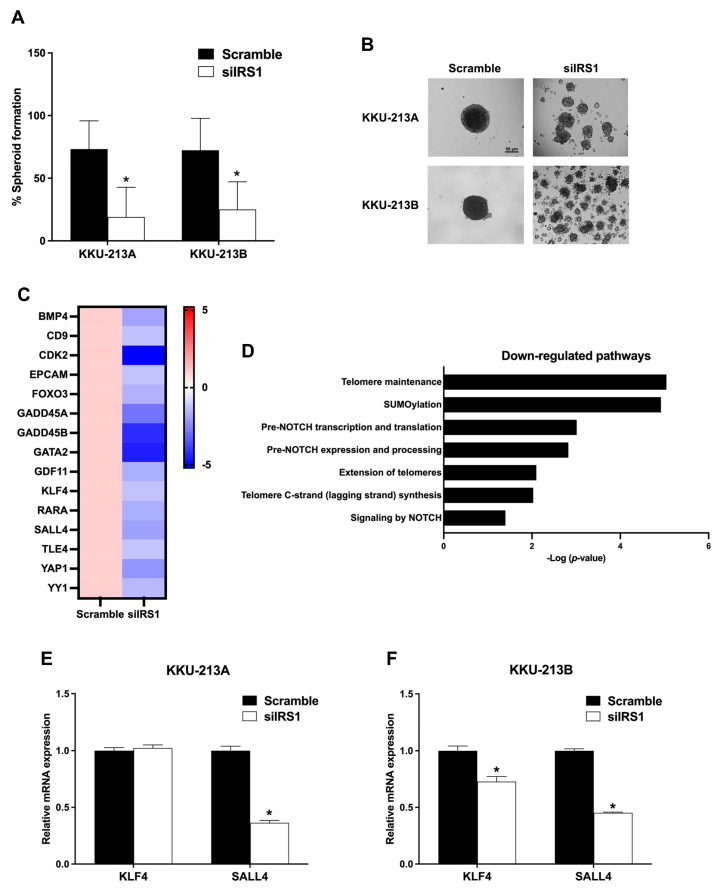
Effect of IRS1 on stem cell properties of CCA cell lines. (**A**) The percentage of spheroid-forming cells in KKU-213A and KKU-213B after IRS1 knockdown; at least 12 spheroids were analyzed in each condition. (**B**) Representative pictures of spheroid formation of KKU-213A and KKU-213B in bright-field microscopy (100× magnification). (**C**) Heatmap represents fold changes of stem cell-related gene expressions in IRS1 knockdown KKU-213B cells investigated by RNA sequencing. (**D**) Reactome pathway demonstrated downregulation of stem cell-related pathways in IRS1 silencing KKU-213B cells. (**E**,**F**) The mRNA levels of stem cell markers (KLF4 and SALL4) in CCA cell lines after IRS1 suppression. * *p*-value < 0.05 compared to scramble.

**Figure 8 ijms-24-02428-f008:**
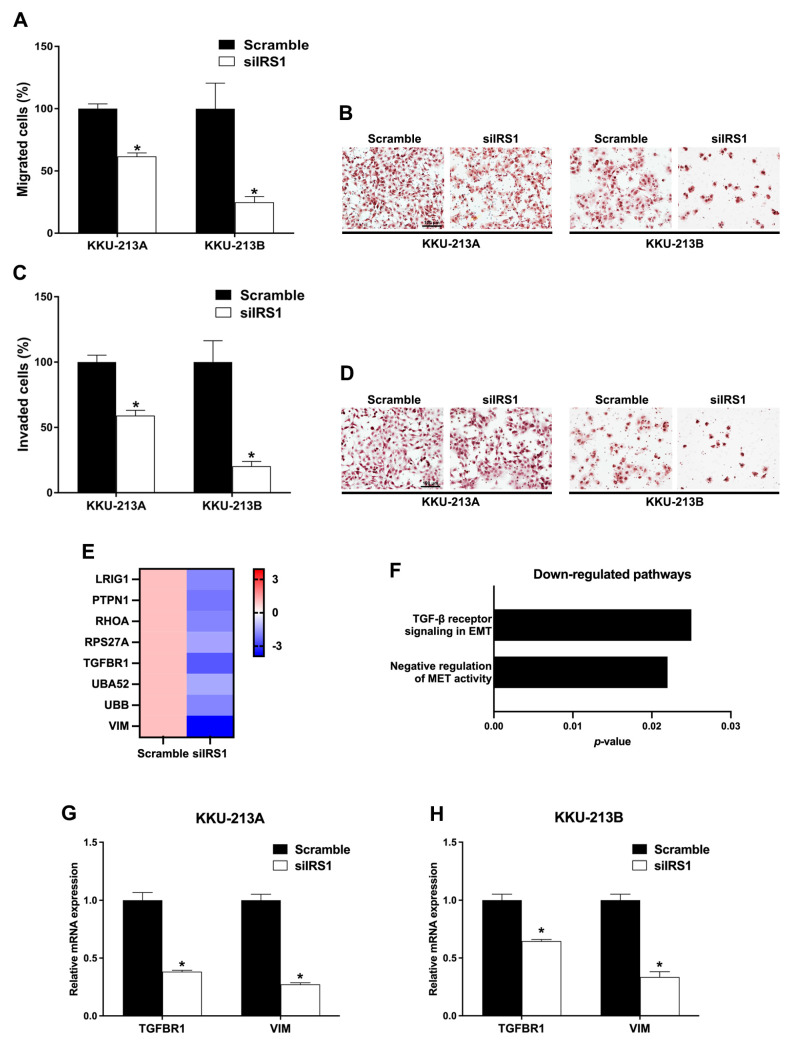
Effect of IRS1 knockdown on cell migration and invasion of CCA cell lines. (**A**) Percentage of migrated cells and (**B**) hematoxylin staining of migrated cells after IRS1 knockdown at 18 h post-migration. (**C**) Quantification of the transwell invasion assay and (**D**) hematoxylin staining of invaded cells after IRS1 knockdown at 18 h post-invasion. (**E**) Heatmap represents fold changes of down-regulated EMT genes investigated by RNA sequencing. (**F**) Down-regulated pathways in IRS1 knockdown KKU-213B cells analyzed by Reactome pathway analysis. (**G**,**H**) Relative mRNA levels of TGFBR1 and VIM in IRS1 knockdown cells. * *p*-value < 0.05 compared with scramble control.

**Figure 9 ijms-24-02428-f009:**
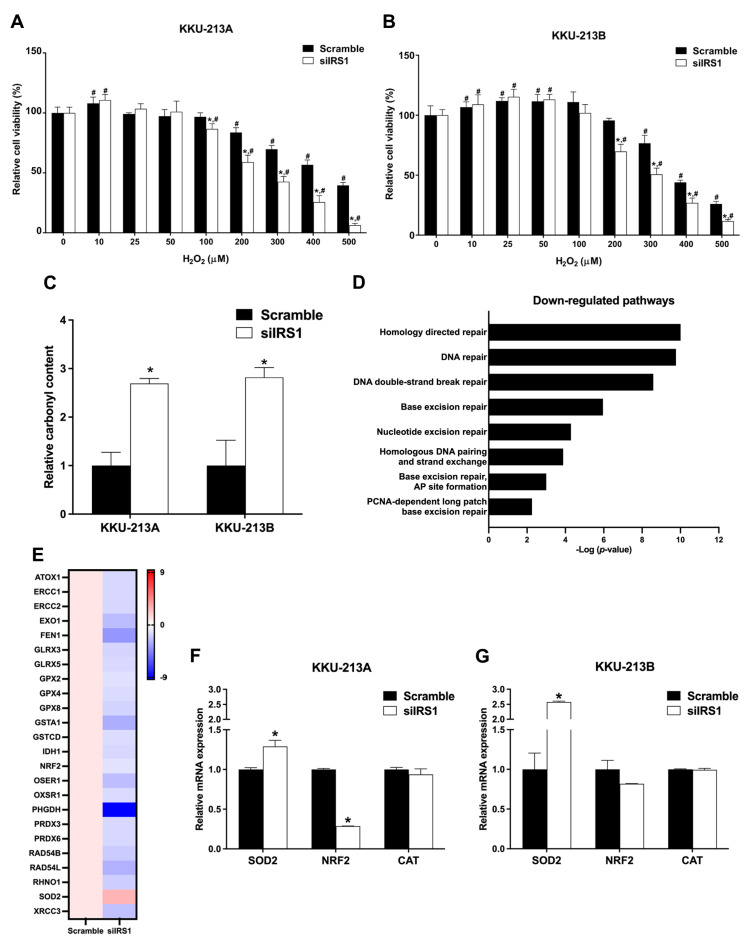
Effect of IRS1 knockdown on oxidative stress-resistant properties of CCA cell lines. (**A**,**B**) Relative cell viability percentage (detected by MTT assay) of KKU-213A and KKU-213B after IRS1 knockdown following 48 h of H_2_O_2_ treatment. (**C**) Bar graph representing relative protein carbonyl content in CCA cell lines after IRS1 knockdown. (**D**) Down-regulated pathways in IRS1 knockdown KKU-213B cells analyzed by Reactome pathway analysis. (**E**) Heatmap representing fold changes of expression of antioxidant genes in IRS1 knockdown KKU-213B cells investigated by RNA sequencing. (**F**,**G**) The mRNA levels of SOD2, NRF2, and CAT in CCA cell lines after siIRS1 transfection. * *p*-value < 0.05 compared to scramble and ^#^
*p*-value < 0.05 compared to 0 μM of H_2_O_2_.

**Figure 10 ijms-24-02428-f010:**
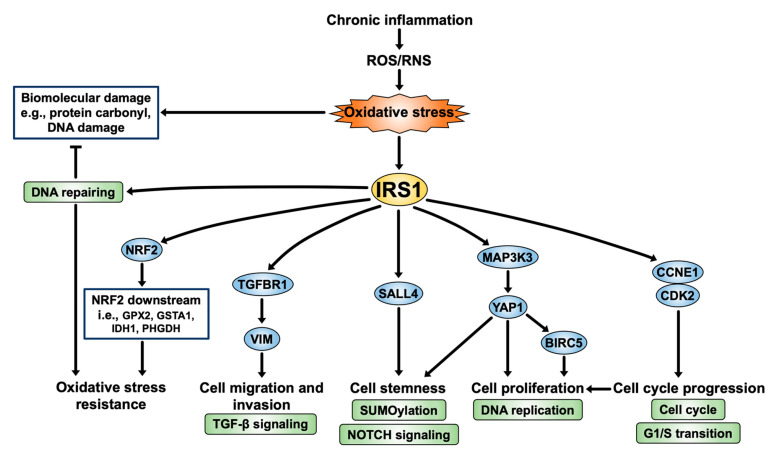
IRS1 signaling pathway: a perspective on CCA progression.

**Table 1 ijms-24-02428-t001:** The association between IRS1 expression patterns and clinicopathological features of CCA patients.

	IRS1 Protein Expression		Statistical Analysis
	Low (*n* = 31)	High (*n* = 53)	
Survival days (Mean ± SD)	480 ± 342	301 ± 218	*p* = 0.007 ^#^
Age (≤57/>57)	15/16	26/27	*p* = 0.953 *
Gender (Male/Female)	21/10	40/13	*p* = 0.443 *
Histology types			
Tubular/Papillary	17/14	32/21	*p* = 0.619 *
Metastasis			
No/Yes	17/14	18/35	*p* = 0.061 *
8-oxodG formation			
Low/High	21/10	16/37	*p* < 0.001 *

* *p*-value was analyzed using Pearson’s chi-square test. ^#^
*p*-value was analyzed using log-rank test.

**Table 2 ijms-24-02428-t002:** The association between combination of IRS1 expression patterns and 8-oxodG formation and clinicopathological features of CCA patients.

	IRS1 and 8-oxodG Protein Expression			Statistical Analysis *
	Low IRS1 and 8-oxodG (*n* = 21)	High IRS1 or 8-oxodG (*n* = 26)	High IRS1 and 8-oxodG (*n* = 37)	
Age (≤57/>57)	10/11	12/14	19/18	*p* = 0.913
Gender (male/female)	16/5	18/8	27/10	*p* = 0.866
Histology types				
Tubular/Papillary	8/13	17/9	24/13	*p* = 0.094
Metastasis				
No/Yes	14/7	8/8	13/24	*p* = 0.026

* *p*-value was analyzed by Pearson’s chi-square test.

## Data Availability

All figures and data are included in the manuscript and Appendix A.

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
