# Peer review of "Overexpression of Insulin Receptor Substrate 1 (IRS1) Relates to Poor Prognosis and Promotes Proliferation, Stemness, Migration, and Oxidative Stress Resistance in Cholangiocarcinoma"

_ijms, 2023, doi:10.3390/ijms24032428_

Round 1
Reviewer 1 Report
In this manuscript authors aimed to determine the effects of oxidative stress induced IRS1 expression on development of CCA.
This study is well-written, interesting and contains original data but I have some concerns about the results and discussion part.
1. Previous studies have shown that oxidative stress induces IRS1 degradation through serine and threonine phosphorylations and overexpression of IRS1 inhibits ROS production
· Archuleta L.T. et al., Free Radical Biology and Medicine, 2009, Oxidant stress-induced loss of IRS-1 and IRS-2 proteins in rat skeletal muscle: Role of p38 MAPK.
· Shih-Hung Chan et al., Journal of Biomedical Science volume 19, Article number: 64 (2012) Insulin receptor substrate-1 prevents autophagy-dependent cell death caused by oxidative stress in mouse NIH/3T3 cells
· R Vinayagamoorthi et al., Journal of Endocrinology, 197,2,2008, Antioxidants preserve redox balance and inhibit c-Jun-N-terminal kinase pathway while improving insulin signaling in fat-fed rats: evidence for the role of oxidative stress on IRS-1 serine phosphorylation and insulin resistance
· Bloch-Damti et al, Diabetologia, 49 (2006) Differential effects of IRS1 phosphorylated on Ser307 or Ser632 in the induction of insulin resistance by oxidative stress
In this study, authors showed the contradictory results. Although, authors showed overexpression of IRS1 and formation of 8-oxoG are correlated with proliferation and decreased survival rates in cancer tissues using IHC, they did not discuss them in this point of view. I think it could be better to confirm IHC results with WB in some samples.
2. In result section, authors mentioned that cell proliferation and migration rates of ox- stress-resistant cells were significantly increased ox-MNK1-L cells. It should be better to show the cell viability and migration results as a supplement.
3. In result section, “However, short-term treatment with H2O2 (0, 10, 25, 50, and 100 µM) for 24, 48 and 72 h had no effect to IRS1 expression levels in MMNK1 and CCA cell lines (data not shown).” In previous studies low dose H2O2 and short term incubations led to decrease IRS1 at protein level. Therefore, authors should show these results as a supplement. We can also see the effects of H2O2 on KKU231A and KU213B but I think this point should be discussed well and clarified.
4. Authors selected KKU-213A and KKU-213B for further experiments but they did not explain why they chose them. Because in western blot IRS1 expression levels are really high in KKU-055 and KKU-023 cells.
5. I am not sure that nuclear IRS1 expression is high in tumor samples. It may be the quality of IHC pictures. Therefore, it should be performed and confirmed by nuclear-cytoplasmic extracts.
6. In siRNA infections, it should be tested by two siRNa construct at least. Although authors showed the decreased expression level of IRS1 in figure, they should explain why they used just one construct.
7. In Figure 4E, in scramble sample of KKU213B, it seems more than one band in WB analyses. It should be repeated using more concentrated WB gel or run more.
8. Authors should show the gatings in flow results in addition to bar graph.
9. In the literature, in addition to IRS1 expression, post-translational modifications of IRS1 have pivotal role in developing cancer (Gorgisen G. et al., 2017, The role of insulin receptor substrate (IRS) proteins in oncogenic transformation) Therefore, IRS1 protein expressions, tyrosine phosphorylation or/and common S/T phosphorylation levels should be shown by WB.
10. KKU-213A and B cells responded different after knocking down of IRS1 esp oxidative stress markers levels. Authors should explain the difference between KKU-213A and B and discussed well.
11. In discussion part, Post-translational modifications of IRS1 and contradictory results should be discussed well.
The study is well organized and authors tried to show the role of oxidative stress induced IRS1 expression in all aspects of CCA development but I think some results should be confirmed by WB analysis. There are many studies about relationship between oxidative stress and IRS1 expression at protein level in the literature and they showed the degradation of IRS1 so authors should support their results using WB and discuss their results well.
Reviewer 2 Report
The manuscript by Kaewlert et. al., attempts to establish a link between insulin receptor substrate 1 (IRS1) and cholangiocarcinoma (CCA). The authors find that oxidative stress upregulates the expression of IRS1 and find a similar upregulation in patients with CCA. The authors also find a correlation between high expression of IRS1 and disease progression. Authors further confirm these findings in immortalized cholangiocyte cell lines and find a correlation between overexpression of IRS1 and cell proliferation. Subsequently, downregulation of IRS1 resulted in inhibition of proliferation, cell migration and invasiveness. Based on these findings authors propose that upregulated IRS1 can be used as a prognosis marker and potential therapeutic target. Most of the experiments conducted are straightforward, however in some instances the presentation must be improved for clarity before the manuscript can be accepted for publication. My comments have been elaborated below.
1) In Figure 1 authors show staining of IRS1 in bile duct from control and high and low expressing CCA patient tissues. The authors should show a higher magnification which is same across all the images. The authors should point to the staining which we are supposed to see and indicate the nucleus and the cytoplasm since authors mention that the staining is seen in both places. A box as used in control tissue should also be used in patient tissues to indicate the region of interest. Secondly in the low expression IRS1, the staining levels seem to be lower than control tissue. Authors should comment on this observation. If the authors still have the tissues, it would be good to also show western blots with appropriate loading controls to show the levels of protein.
2) To test the effects of H2O2 on IRS1 expression in cell lines authors refer to a reference but they should indicate in the text as to how long the treatment was done to see the effects. Authors have subsequently stated that short term treatment even up to 72 hours had no effect. The authors must also comment if the long term H2O2 treatment has any other effects on cell health.
3) The data presented by different methods in Figure 4 seem to be contradictory. Based on immunostaining and western blot data, the cell line KKU-023 has the highest level of IRS1 protein. Why are the mRNA levels then not matching. On the other hand, KKU-213B seems to have similar protein content as the KKU-023 cell line using western blot but seems to have the highest mRNA expression. Authors must comment on this discrepancy in their data. Also, what was the rationale for choosing the last two cell lines KKU-213A and KKU-213B for subsequent experiments? In Figure 4E why are there two bands on the top of each other for IRS1 in scramble siRNA condition for KKU-213B cells?
4) Authors for stemness, proliferation and migration in IRS1 knockdown in a systematic fashion using cell based and transcriptomic analysis. However, the authors don’t test for changes in apoptosis. Authors should show that there is no cell death upon IRS1 downregulation. Another experiment which will strengthen the findings of the authors would be to examine what happens in CCA patients showing high IRS1 expression. The expectation would be a reversal of some or all the phenotypes which are observed in IRS1 knockdown. Such an experiment would establish a strong causal relationship between IRS1 levels and disease progression. Authors also had shown in Table1 that there were some patients which don’t show high IRS1 expression. These would serve as interesting controls. No need to analyze in all patients but may be 3-5 patients with high and low expression of IRS1.
5) What is the difference between migration and invasion. The assay looks similar as described in materials and methods. Also using the assay described it is difficult to account for initial cell density. The authors must do scratch wound assays which should replicate the differences seen with IRS1 knockdown.
Overall, this is a well conducted study in most parts providing interesting links between oxidative stress and inflammation in CCA and IRS1 expression levels. Addressing the above-mentioned experiments will help improve the quality of the findings.
Round 2
Reviewer 1 Report
Dear Authors
Thank you very much for your kindly response to my suggestions and recommendations.
Reviewer 2 Report
Authors have addressed most of my concerns and made an effort to clarify the information in text. It is unfortunate that authors do not have access to the tissue which would have helped for more clarification on some of the points I had raised in the first iteration.
Overall, the study is well conducted, and I recommend it for publication.